# Inactivation of a New Potassium Channel Increases Rifampicin Resistance and Induces Collateral Sensitivity to Hydrophilic Antibiotics in *Mycobacterium smegmatis*

**DOI:** 10.3390/antibiotics11040509

**Published:** 2022-04-12

**Authors:** Thi Thuy Do, Jerónimo Rodríguez-Beltran, Esmeralda Cebrián-Sastre, Alexandro Rodríguez-Rojas, Alfredo Castañeda-García, Jesús Blázquez

**Affiliations:** 1Department of Agriculture, Food and the Marine, Backweston Campus, W23 X3PH Celbridge, Ireland; thuydot82@gmail.com; 2Department of Microbiology, Hospital Universitario Ramón y Cajal, Instituto Ramón y Cajal de Investigación Sanitaria (IRYCIS), 28034 Madrid, Spain; jeronimo.rodriguez.beltran@gmail.com; 3Centro Nacional de Biotecnología-CSIC, Darwin 3, Campus de la Universidad Autónoma de Madrid, 28049 Madrid, Spain; esmeralda.cebrian@cnb.csic.es; 4Internal Medicine—Vetmeduni Vienna, Veterinärplatz 1, 1210 Vienna, Austria; alexandro.rojas@vetmeduni.ac.at

**Keywords:** mycobacteria, potassium channel, rifampicin resistance, collateral susceptibility, ionic balance, isoniazid

## Abstract

Rifampicin is a critical first-line antibiotic for treating mycobacterial infections such as tuberculosis, one of the most serious infectious diseases worldwide. Rifampicin resistance in mycobacteria is mainly caused by mutations in the *rpoB* gene; however, some rifampicin-resistant strains showed no *rpoB* mutations. Therefore, alternative mechanisms must explain this resistance in mycobacteria. In this work, a library of 11,000 *Mycobacterium smegmatis* mc^2^ 155 insertion mutants was explored to search and characterize new rifampicin-resistance determinants. A transposon insertion in the MSMEG_1945 gene modified the growth rate, pH homeostasis and membrane potential in *M. smegmatis*, producing rifampicin resistance and collateral susceptibility to other antitubercular drugs such as isoniazid, ethionamide and aminoglycosides. Our data suggest that the *M. smegmatis* MSMEG_1945 protein is an ion channel, dubbed MchK, essential for maintaining the cellular ionic balance and membrane potential, modulating susceptibility to antimycobacterial agents. The functions of this new gene point once again to potassium homeostasis impairment as a proxy to resistance to rifampicin. This study increases the known repertoire of mycobacterial ion channels involved in drug susceptibility/resistance to antimycobacterial drugs and suggests novel intervention opportunities, highlighting ion channels as druggable pathways.

## 1. Introduction

Tuberculosis (TB), caused by *Mycobacterium tuberculosis* (Mtb) is still the leading cause of death from a single infectious agent. An estimated 10 million people fell ill with TB in 2020, with about 1.4 million deaths globally [1]. Multidrug-resistant TB (MDR-TB), defined as TB resistant to isoniazid (INH) and rifampicin (RIF), and extensively drug-resistant tuberculosis (XDR-TB), resistant to rifampicin and isoniazid plus second-line antibiotics including a fluoroquinolone and either (or both) bedaquiline or linezolid, continue to be a public health threat. Notably, all resistance mechanisms in Mtb are based on chromosomal mutations, and in many cases, Mtb strains have accumulated mutations that confer resistance to multiple drugs.

The low permeability of the mycobacterial hydrophobic cell wall is considered a major reason for intrinsic resistance of mycobacteria to most antibiotics. However, other mechanisms of antibiotic resistance such as efflux pumps, antibiotic degrading/modifying enzymes and target-encoding proteins are well-known [2].

Rifampicin is an effective first-line anti-tuberculosis drug that reduces the duration and failure of antimicrobial chemotherapy for TB. In Mtb, 90–95% of rifampicin-resistance is caused by mutations in the *rpoB* gene, encoding the β subunit of RNA-polymerase [3]. Therefore, 5–10% of Mtb rifampicin resistance must be provided by alternative mechanisms not being fully addressed in the literature [4,5,6].

*M. smegmatis* is a useful surrogate of *M. tuberculosis* to interrogate about the potential to develop resistance, as they share many common mechanisms of resistance, with up to 90% of genetic identity [7]. Indeed, using *M. smegmatis* as a model, we have shown that the inactivation of the potassium uptake regulator TrkA conferred increased resistance to rifampicin [8]. K^+^ interchange is a key physiological process involved in intracellular ionic balance, osmoregulation, pH homeostasis, regulation of protein synthesis, enzyme activation, membrane potential adjustment and electrical signalling of the bacterial cell [9].

In this work, we have characterized a new rifampicin-resistance candidate gene encoding a voltage-gated potassium channel in *M. smegmatis*.

## 2. Results

### 2.1. Identification of the mchK Gene as a Modulator of Rifampicin Resistance

By screening for clones with increased resistance to rifampicin in a library of transposon insertions of *M. smegmatis* mc^2^ 155 [8], we selected a mutant able to grow on agar plates with 20 µg/mL of RIF. The sequence of the transposon insertion point indicated a disruption in the MSMEG_1945 gene at the TA dinucleotide position +1066 (Figure 1a). Analysis by protein blast (BLASTp/NCBI) of the deduced protein sequence showed a polypeptide of 364 amino acids with considerable sequence similarity (73%) with the transmembrane cation channel CglK of *Corynebacterium glutamicum* [10]. The *M. smegmatis* protein also showed an 81% sequence similarity with another putative cation channel from *M. tuberculosis* (Rv3200c) (Figure 1b). The *M. smegmatis* target gene MSMEG_1945 identified in our screening was denominated *mchK* (for mycobacterial channel K^+^). Gene products were denominated MchK_sm_ and MchK_tb_ to define proteins from *M. smegmatis* and *M. tuberculosis*, respectively.

Based on its homology to other K^+^ transporters [10,11,12], the MchK_sm_ protein is predicted to contain two putative domains: an ion transporter channel domain, located inside the membrane, and a regulator of the conductance of potassium ion (RCK) domain exposed to the cytoplasm (Figure 1a). The channel, as in other K^+^ transporters, is putatively formed by a homotetramer of MchK proteins, with each protomer consisting of two transmembrane helices and a pore domain in between them. While the channel domain could mediate the transport of K^+^ inside the cell, the RCK domain is considered to be the cytoplasmic sensor controlling the K^+^ gate [10,11,12]. The sequence TxGYG, inside the channel sequence, is the signature of a selective filter that indicates a specific high conductance of K^+^ ions [10,11,12].

The phenotypic analysis of the *mchK* insertion mutant, isolated from our insertion library, showed that the mutant has an increase in rifampicin resistance. To discard possible polar effects, an in-frame *mchK* knock-out mutant was constructed by deleting the full-length target gene, generating a Δ*mchK* strain (see Section 4). The minimal inhibitory concentrations (MICs) of rifampicin for the wild-type and *mchK* deletion strains were 2 µg/mL and 32 µg/mL (16-fold higher), respectively (Table 1). A plasmid containing the wild-type *mchK* gene (pVV16-*mchK*) rescued the wild-type phenotype in the ∆*mchK* strain (MIC 4 µg/mL). This result demonstrates that the deletion of *mchK* is the key factor contributing to the increase in rifampicin MIC of the *M. smegmatis* ∆*mchK* deletion strain.

### 2.2. Role of the mchK Gene in Collateral Susceptibility of M. smegmatis

Because MchK protein is a putative K^+^ channel with a cytoplasmatic RCK regulator, it may also have a tight relation with the antibiotic susceptibility profile of mycobacteria. To examine this idea, MICs of various antimycobacterial agents against the ∆*mchK* mutant and the wild-type were determined.

While the wild-type strain showed an isoniazid MIC of 128 µg/mL, the ∆*mchK* showed a MIC of 8 µg/mL (Table 1). The mutant was also more susceptible than the wild-type strain to ethionamide (MICs of 64 and 8 µg/mL, respectively), indicating the important contribution of the *mchK* gene in the intrinsic resistance mechanism of *M*. *smegmatis* against these antimicrobials.

The analysis of the antibiotic susceptibility profile showed that the ∆*mchK* mutant is more resistant to hydrophobic antibiotics but more susceptible to hydrophilic ones. The correlation of antibiotic activity with the predicted hydrosolubility coefficients (LogS) are presented in Table 1 (DrugBank; http://www.drugbank.ca; accessed on 1 January 2011). Large hydrophobic antibiotics, such as rifampicin, penetrate cells by passive diffusion. Therefore, reduced permeability to hydrophobic drugs can be responsible for the resistance phenotype of the ∆*mchK* mutant, similar to the previously described ∆*trkA* variant [8]. In this sense, the MIC of novobiocin, another highly hydrophobic drug, increased 2-fold in the ∆*mchK* mutant compared with that of the wild-type. The MICs of less hydrophobic drugs with a large apolar core, such as fluoroquinolones (ciprofloxacin and ofloxacin), were similar between the wild-type and the mutant strains (Table 1). By contrast, aminoglycosides (streptomycin, amikacin and kanamycin), positive charged highly hydrophilic antimycobacterial drugs, showed increased activity against the mutant derivative, similar to isoniazid.

The viability of the ∆*mchK* strain in the presence of rifampicin, ciprofloxacin and isoniazid at different concentrations was also analysed. As expected, viable counts of the ∆*mchK* variant under different concentrations of rifampicin were significantly greater than those of the wild-type (Figure 2a). This result explains the higher MIC values for the ∆*mchK* strain shown in Table 1. Although the mutant and the wild-type strains show the same ciprofloxacin MIC, the mutant displayed higher viability responding to increasing concentrations of ciprofloxacin than the wild-type (Figure 2b). By contrast, the viability of the mutant ∆*mchK* strain is significantly decreased when the concentration of isoniazid increases (Figure 2c). The wild-type viability also decreased with increasing isoniazid concentrations but still remained at a higher level in comparison with the mutant. This result confirmed the difference between isoniazid MICs of the wild-type and the ∆*mchK* mutant. In summary, the reaction of ∆*mchK* mutant to different antibiotics demonstrates the important role of the *mchK* gene in controlling the susceptibility to both hydrophobic and hydrophilic antibiotics, including first-line (rifampicin and isoniazid) and second-line (ciprofloxacin) drugs.

### 2.3. Effect of mchK Deletion on Growth and K^+^ Requirement

To study the effect of *mchK* deletion on bacterial growth, the growth ability of wild-type and ∆*mchK* mutant were assessed by measuring optical density at the mid-exponential phase (after 12 h of growth). Deletion of *mchK* gene imposed a fitness cost reflected in a notably lower growth than that of the wild-type in 7H9 Middlebrook broth (Figure 3a). When K^+^ (in the form of KCl) was added to the medium, the growth of the mutant approached that of the wild-type as long as concentrations of KCl increased, supporting its role as an uptake transport of K^+^ inside the cells. These results suggest that the ∆*mchK* mutant presents a reduced K^+^ uptake due to a lack of MchK-mediated transport and that it requires an additional K^+^ supply to counterbalance its growth defect.

### 2.4. The mchK Gene and pH Homeostasis

Potassium and acid–base balance are known to be interrelated. At low internal pH, cells exchange intracellular protons by extracellular K^+^ to maintain the internal pH [13]. Therefore, loss of the K^+^ uptake by disruption of K^+^ transport should increase the bacterial susceptibility to acidic conditions. To examine the activity of the *mchK* gene in the regulation of internal pH, the ∆*mchK* mutant was grown in liquid medium at different pHs, ranging from 5 to 8. The mutant grew at a lower rate than the wild-type at neutral pH (Figure 3b). At acidic pH, the growth of the mutant was severely impaired in comparison with that of the wild-type. However, both wild-type and mutant strains showed the same growth capacity under alkaline pH. In addition, the growth defect at pH 5.5 was compensated with high concentrations of K^+^ in a dose-dependent manner (Figure 3c). These results indicate that the *mchK* gene activity is necessary for pH homeostasis.

### 2.5. The mchK Gene and Membrane Potential

Maintenance of the optimal transport of K^+^ is required for an optimal membrane potential value, being the main contributor to proton motive force (PMF) when cells grow in a neutral environment [14]. To study the contribution of *mchK* to membrane potential maintenance, we measured electrochemical membrane potentials across the membrane by monitoring the quenching of the rhodamine 123 fluorescence. Cells with high membrane potential—that is, with increased negative charge inside—tend to accumulate the cationic probe rhodamine 123. Inside cells, the probe fluorescence is quenched, leading to a decrease in fluorescence signal which is quenched [15]. The fluorescence decay is, thus, proportional to the electrical membrane potential. Cultures of the ∆*mchK* mutant showed lower fluorescence levels than those of the wild-type strain (Figure 4)—i.e., quenching of rhodamine 123 and consequent fluorescence decay—is higher in the mutant. This indicated that the inactivation of *mchK* produces an increase in the negative charge inside the cell with higher membrane potential. Notably, preincubation of the cells with K^+^ (100 mM KCl) for 30 min depolarises the bacterial membrane and counteracts the higher negative charge inside the ∆*mchK* cells (Figure 4). This result may explain the observed multidrug susceptibility profile of the ∆*mchK* mutant.

## 3. Discussion

The vast majority of rifampicin-resistant strains arise through mutations in the *rpoB* gene in mycobacteria. Some rifampicin-resistant strains, however, show no *rpoB* mutations, indicating the existence of alternative mechanisms responsible for RIF resistance in mycobacteria. In this work, we identified and characterised a *M. smegmatis* mutant displaying a high level of rifampicin resistance with a 16-fold increase in the MIC when compared with that of the wild-type. Disruption of this gene is the solely responsible for the observed phenotype, as demonstrated by complementation experiments.

### 3.1. MchK as a Putative K^+^ Transport Protein

The mutant carries an insertion in the gene MSMEG_1945, encoding a protein with high similarity to the ion channel CglK from *C. glutamicum* [10], which is relevant for major physiological processes, such as the activity of the respiratory chain, the maintenance of the internal pH and the adjustment of the membrane potential [10]. The predicted *M. smegmatis* protein shows, as well as CglK, an ion channel domain in the membrane, with two transmembrane-spanning regions plus a pore with a K^+^ selectivity filter and an RCK regulator domain exposed to the cytoplasm. The *M. smegmatis* gene has been denominated *mchK*, as it constitutes a new mycobacterial member of the K^+^ ion channel family.

Maintenance of high internal K^+^ concentration is a key factor for bacterial cell survival [16] and is also essential for a variety of processes, such as maintenance of osmotic pressure, activation of enzymes, stress response, gene expression and regulation of cytoplasmic pH [10,16,17,18,19]. The ∆*mchK* mutant requires additional amounts of K^+^, in a dose-dependent manner to grow at the same rate as the wild-type. This result provides physiological evidence that MchK functions as a potassium channel in *M. smegmatis*, promoting the uptake of K^+^ inside the cells.

### 3.2. Potassium Transport via MchK Is Necessary for M smegmatis Growth at Acidic pH

The presence of K^+^ was shown to be necessary for the maintenance of a neutral internal pH for a number of bacteria, including *E coli* [20], *Lactococcus lactis* [13] and *Streptococcus mutans* [21] and *C. glutamicum* [10]. The ∆*mchK* mutant grew similarly to the wild-type under the alkaline pH condition, but its growth was significantly impaired at low pH values. At pH 5.5, additional K^+^ was necessary to support the growth of the mutant at a level similar to that of the wild-type, indicating that its inactivation impairs the K^+^ transport (at least in part), conferring a higher susceptibility to low pH that could only be counterbalanced by high extracellular K^+^. Therefore, the *mchK* gene activity is necessary for pH homeostasis, as it compensates the descent of intracellular pH by promoting the uptake of K^+^.

### 3.3. Role of MchK in Charge Balance and Membrane Potential

Proton motive force (PMF) is an electrochemical ion gradient across the membrane, contributed by two factors: electric potential (Δψ, inside negative) and a chemical gradient (ΔpH, inside alkaline) [22]. In *E. coli*, K^+^ causes the interconversion between the components of PMF. The experiments with K^+^ transport mutants of *E. coli* indicated that bacteria must have a functional K^+^ transport system to maintain that interchange. The inward movement of K^+^ causes the depolarisation of bacterial membrane. The defect of K^+^ retention via the disruption of K^+^ uptake resulted in an increase in membrane Δψ [14]. Here, we found that the mutant lacking MchK protein showed an increase in Δψ, which can be compensated for by the addition of K^+^, supporting its role as a modulator of membrane potential.

### 3.4. MchK and Antibiotic Susceptibility

The ion channels have been considered as a major class of drug targets due to their essential contributions to numerous fundamental physiological processes [12,23]. These proteins are important in the development of antibiotic resistance in bacteria. Indeed, the K^+^ uptake system Trk, including TrkA, modulates antibiotic resistance profile in many bacteria such as *E. coli* [24], *Pseudomonas aeruginosa* [25] and *M. smegmatis* [8]. We found that the deletion of the *mchK* gene in *M. smegmatis* resulted in an increased resistance to hydrophobic/lipophilic antibiotics such as rifampicin, novobiocin and fluoroquinolones. However, it was also associated with a relevant collateral susceptibility to hydrophilic antibiotics, such as isoniazid and aminoglycosides. The ∆*mchK* mutant has a hyperpolarized membrane with more negative charge inside the cells, due to a blocked uptake of potassium. Thus, a high Δψ may help the penetration of positively charged compounds across the membrane and reduce the diffusion of hydrophobic compounds such as novobiocin and rifampicin. In other Gram-positive bacteria such as *Staphylococcus aureus*, potassium transporters play a role in both pathogenesis and antimicrobial resistance, suggesting, thus, possible secondary druggable pathways [26].

The antibiotic susceptibility profile of bacteria also depends on other factors, including porins and drug efflux pumps [27]. For instance, the ABC-type multidrug efflux pumps can interact with the Trk system in *E. coli* [28]. Moreover, the membrane potential and intracellular pH are important factors that regulate the activity of the drug efflux pumps depending on PMF in prokaryotes [29,30]. The K^+^ transport could also have effects on the function and/or regulation of drug efflux pumps. Therefore, the effects of *mchK* inactivation on *M. smegmatis* antibiotic susceptibility could also be attributed, at least in part, to a differential regulation of other genes under osmotic stress, including efflux pumps and/or porins. The relationship between the K^+^ transport systems and the function of drug efflux pumps deserves to be studied in the future to identify additional mechanisms of drug resistance in mycobacteria [31].

In conclusion, the *M. smegmatis* MchK protein appears to be a potassium channel that modulates susceptibility to antimycobacterial agents, including rifampicin resistance. This study increases the repertoire of ion channels involved in drug susceptibility/resistance to antimycobacterial drugs. Our results support the possibility of using potassium channels as a secondary target for enhancing the activity of hydrophilic antitubercular antibiotics, at least in MDR strains.

## 4. Materials and Methods

### 4.1. Bacteria, Media and Growth Conditions

*M. smegmatis* strain mc^2^ 155 and its derivative mutants were grown at 37 °C in Middlebrook 7H9 broth or Middlebrook 7H10 agar containing 0.5% glycerol and 0.05% Tween 80. When appropriate, culture media were supplemented with final 25 µg/mL of kanamycin or 50 µg/mL of hygromycin B for selection purposes. Genetic manipulations were performed in the strain *Escherichia coli* DH5α, cultured at 37 °C in Lysogeny broth (LB) medium or LB agar containing 50 µg/mL of kanamycin or 100 µg/mL of hygromycin when appropriate.

### 4.2. Minimal Inhibitory Concentration (MIC) Determination and Survival Rates to Antibiotics

The MICs for the *M. smegmatis* wild-type strain and its derivative mutants were determined by agar dilution method with Middlebrook 7H10 agar in triplicate. Bacteria were grown in Middlebrook 7H9 broth until the mid-logarithmic phase and then diluted to a final inoculum of 10^6^ cfu/mL. Approximately 10^4^ viable cells were plated on Middlebrook 7H10 agar plates with or without the corresponding antibiotics. The plates were incubated at 37 °C for 5 days. The MIC was defined as the lowest concentration of the antimicrobial that inhibited the visible growth of *M. smegmatis*. To measure survival rates in the presence of different concentrations of rifampicin (1, 2, 4 and 8 µg/mL), isoniazid (0.04, 0.08, 0.16 and 0.32 µg/mL) and ciprofloxacin (16, 32, 64 and 128 µg/mL), bacteria were plated in triplicate on Middlebrook 7H10 agar plates with or without indicated antibiotic concentrations. Survival rates were determined as following:$$Suvival\;rates\;\left(\%\right)=\frac{Surviving\;cells}{Total\;cells}\;\times \;100$$
where surviving cells are those growing on antibiotic-containing plates, and total cells are cells growing on antibiotic-free plates.

### 4.3. Screening of M. smegmatis φMycoMarT7 Insertion Library and Identification of the Insertion Site

The rifampicin-resistant mutants were selected and identified through the screening of a *M. smegmatis* mutant library of random insertions of the transposon φMycoMarT7, as previously described [8]. This library contains 11,000 transposition mutants. Each mutant was replicated onto Middlebrook 7H10 agar plates supplemented with rifampicin at different concentrations (5, 10 or 20 µg/mL). Rifampicin-resistant candidates were regrown on selected plates to confirm their ability to grow in the presence of rifampicin. The rifampicin MIC of selected mutants were determined by the broth microdilution method as stated above.

Sequencing of transposon-disrupted DNA in *M. smegmatis* was conducted as described previously [8]. Briefly, the whole genomic DNA from the resistant isolates was purified, digested with BamHI and religated with T4 ligase. Self-ligated circular DNA originated a plasmid carrying the entire transposon and the flanking chromosomal DNA adjacent to the insertion site. The ligation was electroporated into *E. coli* DH5α λ*pir116* and selected on LB plates with 50 µg/mL of kanamycin. Colonies were subcultured and their plasmids purified and sequenced using the transposon-specific primer 5′-CCCGAAAAGTGCCACCTAAATTGTAAGCG-3′, which hybridizes nearby the junction of transposon/chromosome junction. The DNA sequences adjacent to the insertion site were compared with the *M. smegmatis* wild-type genome sequence by BLASTn (NCBI) to identify the target gene.

### 4.4. Generation of an In-Frame mchK Knockout Mutant Strain

The *M. smegmatis* in-frame *mchK* deletion was obtained as described previously [8]. Briefly, two fragments (~1000 base pairs each), corresponding to the up- and downstream sequence of the *mchK* gene, were amplified by PCR. The oligonucleotides

5′-GACAAAGCTTCGGCCACCGCGCGTGCGGTT-3′ and 5′-GCGAGGTACCATTCGAGTCGATGGCGGCCA-3′ were used for amplification of the upstream part, while 5′-GGCCGGTACCCGGCTCCTGTACATCCGCAG-3′ and 5′-CCAAGCGCGGCCGCCCTTATCCCGAGAATGGGCCAC-3′ corresponded to the downstream region. Both fragments were cloned in-frame into the suicide p2NIL vector [32]. The constructed plasmid was digested with PacI to insert a *hyg sacB lacZ* cassette from pGOAL19 [32] and verified by sequencing. The plasmid with the in-frame deletion of *mchK*, termed p2NIL-Δ*mchK*, was introduced into *M. smegmatis* mc^2^ 155 and plated on Middlebrook 7H10 agar supplemented with kanamycin (25 µg/mL) and hygromycin (50 µg/mL). Once single-crossover clones were obtained, they were grown in Middlebrook 7H9 broth without antibiotics to allow a second crossover event. Finally, the cultures were diluted and counterselected on 7H10 plates containing 10% sucrose and 100 µg/mL X-gal. Candidate colonies were tested for kanamycin and hygromycin sensitivity and analysed by PCR to confirm the unmarked deletion of *mchK*.

### 4.5. Complementation of the ΔmchK Mutant Strain

A plasmid carrying the wild-type *mchK* gene was constructed for complementation studies. Briefly, the *mchK* gene was amplified by PCR from *M. smegmatis* mc^2^ 155 wild-type genomic DNA using the forward and reverse primers 5′-GGATGACATATGGCTAAAGGCAGGTTACGG-3′ and 5′- CGGTAAGCTTTCATCGTTCGGCGTCCGCAC-3′, respectively. The PCR products were digested with NdeI and HindIII and cloned into the pVV16 vector [33] to generate the complementation plasmid. The resulting plasmid, pVV16-*mchK*, was introduced into the *M. smegmatis* Δ*mchK* mutant by electroporation. Transformants were selected on Middlebrook 7H10 agar supplemented with kanamycin (25 µg/mL) and hygromycin (50 µg/mL) and verified by PCR.

### 4.6. Membrane Potential Assay

Membrane potential assays were carried out with *M. smegmatis* wild-type and the mutant strain ∆*mchK* as described previously [8]. When bacteria reached the late logarithmic phase, the cultures were washed and adjusted to roughly the same number of cells (~10^7^ cfu/mL). Cell membrane potential was determined using the fluorescent probe rhodamine 123 (Sigma), a lipophilic cationic molecule that enters into bacterial cells in direct proportion to the strength of electric potential. Inside the cells, the probe fluorescence is quenched, leading to a decrease in the fluorescence signal. Rhodamine 123 was added to samples to a final concentration of 0.5 µg/mL. Cells were pre-incubated for 30 min with or without 100 mM KCl, and fluorescence was measured after 10 min of rhodamine 123 addition to evaluate the remaining fluorescence values in each sample using a Tecan infinite F200 multiplate reader (480 nm excitation, 530 nm emission). The fluorescence data were normalized to the optical density of cultures.

## Figures and Tables

**Figure 1 antibiotics-11-00509-f001:**
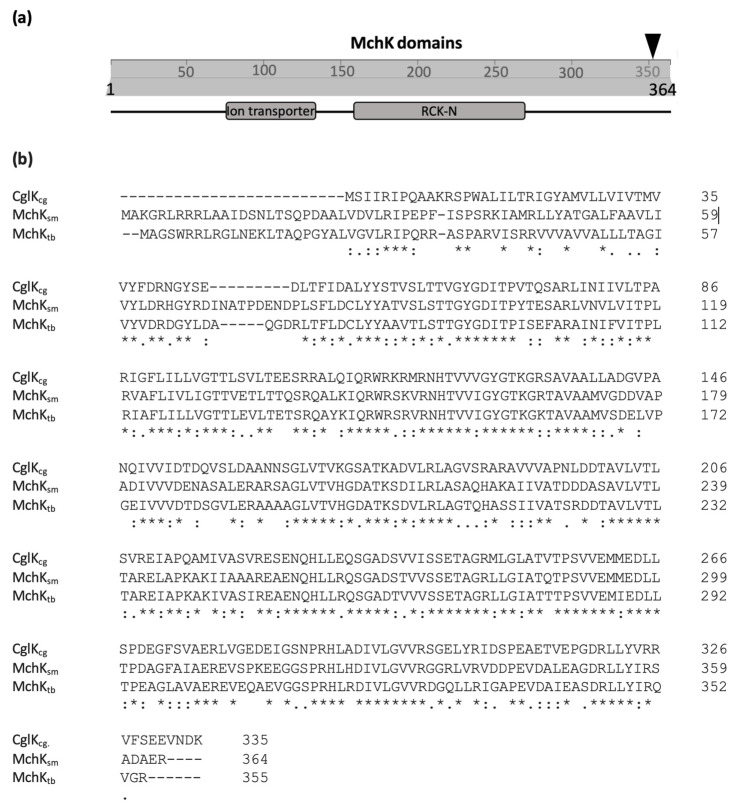
(**a**) Predicted domain structure MchK. The arrow indicates the transposon insertion position at TA dinucleotide position (+1066) of the gene. The predicted transmembrane ion transporter channel domain is located between amino acids 79 and 136 and the RCK domain (cytoplasmic) is between amino acids 157 and 272 (https://www.ncbi.nlm.nih.gov/Structure/cdd/wrpsb.cgi?INPUT_TYPE=live&SEQUENCE=WP_003893325.1; accessed on 10 February 2022). (**b**) Alignment of three protein sequences: CglK_cg_ from *C. glutamicum*, MchK_sm_ from *M. smegmatis* and MchK_tb_ from *M. tuberculosis*. (:) indicates conservation between groups of strongly similar properties. (.) indicates conservation between groups of weakly similar properties. The asterisk (*) indicates identical amino acids from the three protein sequences.

**Figure 2 antibiotics-11-00509-f002:**
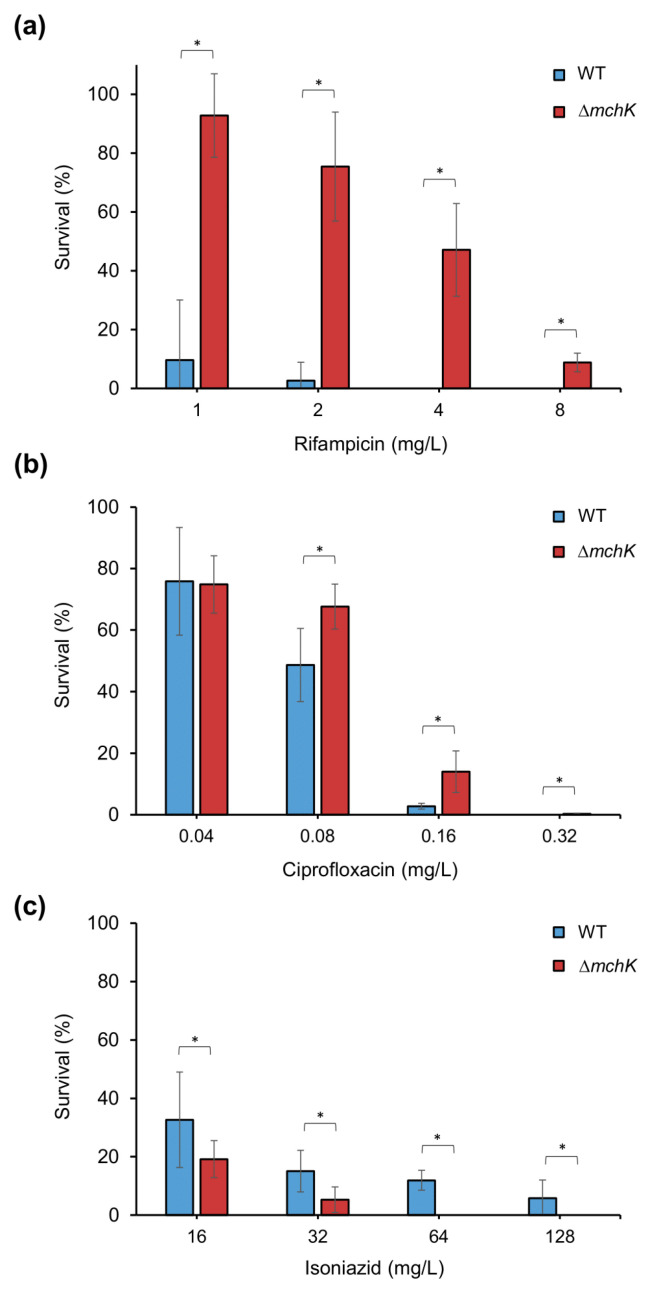
Survival (%) of *M. smegmatis* mc^2^ 155 wild-type (blue bars) and the ∆*mchK* mutant (red bars) under different antibiotic concentrations. Percentages of survivors of the wild-type and the ∆*mchK* mutant to rifampicin (**a**), ciprofloxacin (**b**) and isoniazid (**c**) at the indicated antibiotic concentrations are shown. Error bars indicate 95% CI (confidence intervals); (*) *p* < 0.05 (*t*-test).

**Figure 3 antibiotics-11-00509-f003:**
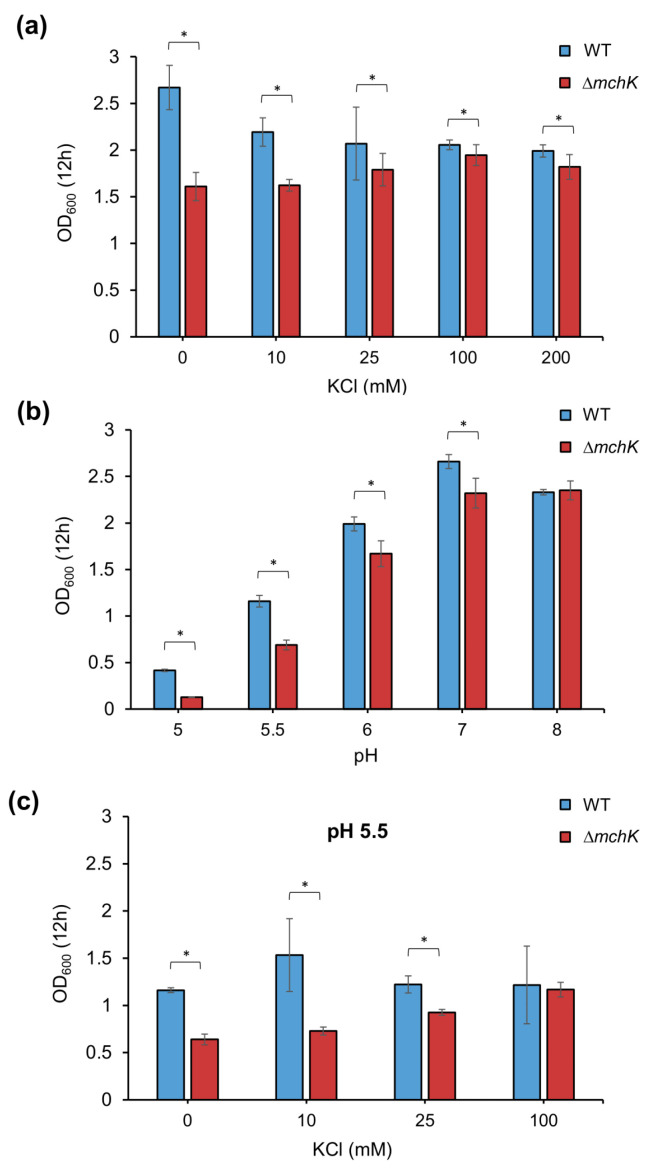
K^+^ effect on the growth of mc^2^ 155 and its ∆*mchK* derivative under different conditions. (**a**) Effect of K^+^ addition (10–200 mM KCl) on the growth of mc^2^ 155 and ∆*mchK* at neutral pH. The ability of high K^+^ concentrations to restore the growth of the ∆*mchK* mutant was measured in cultures at mid-exponential phase (12 h). (**b**) Effect of extracellular pH (5–8) on the growth of mc^2^ 155 and the ∆*mchK* mutant in Middlebrook 7H9 liquid medium after 12 h of incubation (initial OD_600_ 0.05). (**c**) Effect of K^+^ addition (10–100 mM KCl) on the growth of mc^2^ 155 and ∆*mchK* at pH 5.5. The OD_600_ was measured after 12 h of incubation at 37 °C. Error bars indicate 95% CI (confidence intervals); (*) *p* < 0.05 (*t*-test).

**Figure 4 antibiotics-11-00509-f004:**
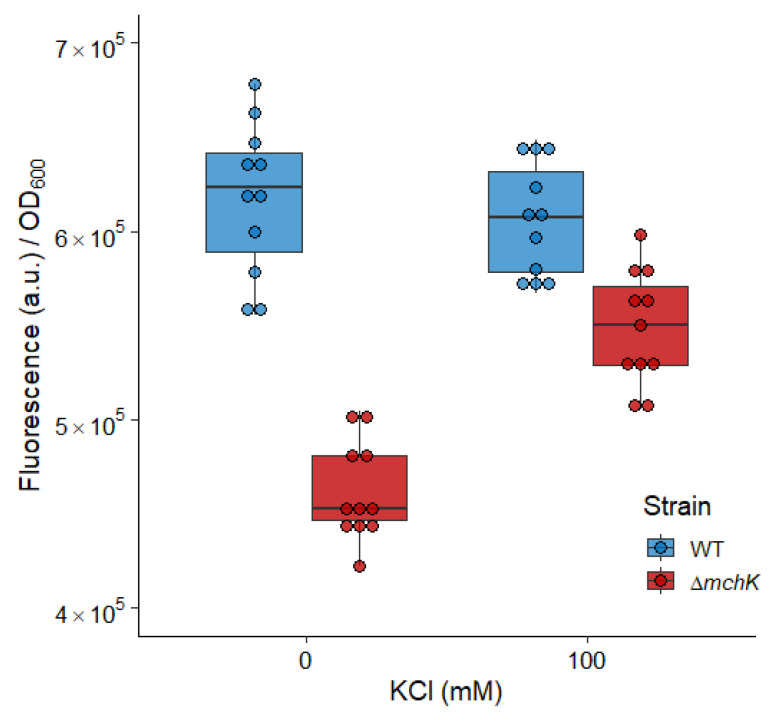
Membrane potential of mc^2^ 155 and ∆*mchK* and effect of K^+^ addition. Cells were pre-incubated for 30 min without or with KCl (100 mM). Fluorescence was measured after 10 min of incubation with rhodamine 123. Dots represent the fluorescence data normalized to the OD_600_ of the cells of both wild-type (blue) and mutant (red) strains. Boxplots represent median values (horizontal line in the box). The depth of the box represents the interquartile range (50% of the data), and the whiskers extend to 1.5 times the interquartile range.

**Table 1 antibiotics-11-00509-t001:** MICs of different antibiotics for *M. smegmatis* wild-type and its ∆*mchK* derivative. Antibiotics are ordered by solubility values from hydrophobic (up) to hydrophilic (down).

			MICs (µg/mL)	
Antibiotic	Molecular Weight	LogS ^a^	mc^2^ 155	∆*mchK*
**novobiocin**	612	−4.80	32	64
**rifampicin**	822	−4.09	2	32
**ofloxacin**	361	−2.40	0.32	0.32
**ciprofloxacin**	331	−2.39	0.32	0.32
**ethionamid**	166	−2.30	64	8
**ethambutol**	204	−1.43	2	2
**amikacin**	585	−1.07	0.8	0.4
**streptomycin**	581	−0.96	0.8	0.2
**kanamycin**	484	−0.72	3.2	0.8
**isoniazid**	137	−0.59	128	8

^a^ Predicted hydrosolubility (DrugBank; http://www.drugbank.ca; accessed on 1 January 2011).

## Data Availability

Not applicable.

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
