# Peer review of "Inactivation of a New Potassium Channel Increases Rifampicin Resistance and Induces Collateral Sensitivity to Hydrophilic Antibiotics in Mycobacterium smegmatis"

_antibiotics, 2022, doi:10.3390/antibiotics11040509_

Round 1
Reviewer 1 Report
This is an interesting and very important paper that widen our outlook on the problems of the drug resistance in mycobacteria, a group of organisms comprising harmful human pathogens such as Mycobacterium tuberculosis. The existence of pathogens resistant to multiple antibiotics is a real challenge in treating infectious diseases, and investigation of the nature of multidrug resistance is of high importance. The authors studied the nature of mutations (distinct from the well-known rpoB) conferring rifampicin resistance in Mycobacterium smegmatis, a useful surrogate of M. tuberculosis. Using an approach based on transposon insertions, the authors found that the product of MSMEG_1945 gene encoding a putative voltage-gated potassium channel (named here MchK) may serve as a modulator of rifampicin resistance. Moreover, the mchK null-mutant appeared to be more resistant to hydrophobic antibiotics while more susceptible to hydrophilic ones, due to disruption of K+-transport across the membrane. In addition, the authors studied the effect of the mchK deletion on cell growth, K+-requirement and pH homeostasis, and the results obtained supported the role of MchK in transport of K+ inside the cell. The finding of a new mutation in mycobacteria indicates the existence of alternative ways to confer rifampicin-resistance, although the exact mechanism remains unknown. The manuscript is well written, the results are clearly described and properly discussed.
The main point to be addressed is the Table 1 (page 4) where the authors compare the susceptibility of WT and the mchK mutant strains to different antibiotics, and its description in the text (lines 110-114). The authors should carefully check MICs for ethionamide in the Table (64 and 8 mkg/ml for wt and mutant cells, respectively) and in the text (32 and 128, respectively). The discrepancy should be correct!
Author Response
Response: Thank you for the appropriate comment. MICs of ethionamide indicated in the text were due to a mistake. Now, the text has been modified.
Reviewer 2 Report
This is an interesting work that relates antibiotic resistance in M smegmatis to a putative potassium transporter gene product MchK. The results are interesting wherein deletion of this gene results in increased rifampicin resistance. The findings are useful to address non-rpoB rifampicin resistance in TB, and provides additional clues for therapeutic strategies. While the findings are interesting, the discussion of the findings should include additional perspectives.
Additionally, the findings of the paper also opens up a reasonable angle of enquiry that is feasible for the authors to perform. An experiment is suggested (Comment #2 below), that will add value to the paper and make it truly worthy for publication. Thus, this paper is recommended for major revision (including the additional experiment) before publication.
Comment #1
Line 115-127
Large hydrophobic antibiotics, such as rifampicin, penetrate cells by passive diffusion. Therefore, reduced permeability to hydrophobic drugs can be responsible for the resistance phenotype of the ∆mchK mutant, similar to the previously described ∆trkA variant 8. 8. In this sense, the MIC of novobiocin, another highly hydrophobic drug, increased 2-fold in the ∆mchK mutant compared with that of the wild-type. The MICs of less hydrophobic drugs with a large apolar core, such as fluoroquinolones (ciprofloxacin and ofloxacin), were similar between the wild-type and the mutant strains (Table 1). By contrast, aminoglycosides (streptomycin, amikacin and kanamycin), positive charged highly hydrophilic antimycobacterial drugs, showed increased activity against the mutant derivative, similar to isoniazid.
This is one of the key findings in the paper. The conclusion is that rifampicin resistance is correlated to reduced permeability to hydrophobic drugs in the ∆mchK (and similarly for trkA in the cited reference also by the same authors). I would like to raise the question whether this controls for all variables. Many multi-drug resistance genes (including efflux transporters and porins are upregulated under osmotic stress. Even a functional correlation between osmotic stress regulators and MDR has been suggested in literature (e.g (https://www.ncbi.nlm.nih.gov/pmc/articles/PMC3910827/).
That variable is not controlled in the experiment and the expression levels of various drug resistance genes is not determined in this study. While the findings seem to be correct and relevant, the discussion and inferences need to consider the additional nuances of the drug resistance. The alternative hypothesis is that overall “poor health” (also indicated by lower growth compared to WT) caused by the ∆mchK mutation in general reduces the MICs of all antibiotics. However upregulation of certain genes (MDR efflux pumps, porins or specific drug resistance genes) contributes to antibiotic resistance in specific antibiotics (such as rifampicin in this case). It is of great interest to see which genes are upregulated in WT vs ∆mchK backgrounds. If such experiments are within the scope of this investigation it is highly recommended that these experiments are performed and they would really add crucial information to our understanding of the role of osmotic stress in drug resistance. If these are outside the scope of the current manuscript the discussion should be provided with more nuance considering these factors.
Comment #2
The second consideration is that there is a confusing disconnect between the first and second parts of the paper as it is written. The first part predominantly deals with drug resistance and the second part deals with characterizing mchK and its role in monovalent cation and pH balance. Within the scope of this journal the first part is more interesting.
The finding that increasing [K+] poses one very obvious experiment perhaps within the scope of this work is the impact of [K+] on the MIC of rifampicin for WT vs ∆mchK. The findings in the second part gain relevance if such an experiment is included. I recommend that the authors perform this experiment.
Otherwise the paper is well written and additional minor remarks are below.
1. Line 48
failure of chemotherapy treatment for TB.
Change to “antimicrobial chemotherapy” for TB, in order to disambiguate from cancer chemotherapy.
2. Line 71-72
The M. smegmatis target gene MSMEG_1945 identified in our screening was denominated mchK (for mycobacterial channel-K+).
Please also additionally denominate gene products MchKsm and MchKtb in the text.
3. Line 75-81
The channel is formed by a homotetramer of MchK proteins, with each protomer consisting of two transmembrane helices and a pore domain in between them. While the channel domain could mediate the transport of K+ inside the cell, the RCK domain is considered to be the cytoplasmic sensor controlling the K+ gate 12–14 . This channel contains a selective filter inside the pore with the signature sequence TxGYG that indicates a specific high conductance of K+ ions.
These lines should be rewritten to reflect the fact that the authors are hypothesizing the mechanism and structure based on other K+ transporters. As written, it reads as though the cited references (12-14) are based on studies on the “MchK” protein.
4. Line 95
Provide abbreviation expansion for MIC
5. Line 286-296
Methods - Please provide antibiotic concentrations used.
Author Response
Comment #1
Line 115-127
Large hydrophobic antibiotics, such as rifampicin, penetrate cells by passive diffusion. Therefore, reduced permeability to hydrophobic drugs can be responsible for the resistance phenotype of the ∆mchK mutant, similar to the previously described ∆trkA variant 8. 8. In this sense, the MIC of novobiocin, another highly hydrophobic drug, increased 2-fold in the ∆mchK mutant compared with that of the wild-type. The MICs of less hydrophobic drugs with a large apolar core, such as fluoroquinolones (ciprofloxacin and ofloxacin), were similar between the wild-type and the mutant strains (Table 1). By contrast, aminoglycosides (streptomycin, amikacin and kanamycin), positive charged highly hydrophilic antimycobacterial drugs, showed increased activity against the mutant derivative, similar to isoniazid.
This is one of the key findings in the paper. The conclusion is that rifampicin resistance is correlated to reduced permeability to hydrophobic drugs in the ∆mchK (and similarly for trkA in the cited reference also by the same authors). I would like to raise the question whether this controls for all variables. Many multi-drug resistance genes (including efflux transporters and porins are upregulated under osmotic stress. Even a functional correlation between osmotic stress regulators and MDR has been suggested in literature (e.g (https://www.ncbi.nlm.nih.gov/pmc/articles/PMC3910827/).
That variable is not controlled in the experiment and the expression levels of various drug resistance genes is not determined in this study. While the findings seem to be correct and relevant, the discussion and inferences need to consider the additional nuances of the drug resistance. The alternative hypothesis is that overall “poor health” (also indicated by lower growth compared to WT) caused by the ∆mchK mutation in general reduces the MICs of all antibiotics. However upregulation of certain genes (MDR efflux pumps, porins or specific drug resistance genes) contributes to antibiotic resistance in specific antibiotics (such as rifampicin in this case). It is of great interest to see which genes are upregulated in WT vs ∆mchK backgrounds. If such experiments are within the scope of this investigation it is highly recommended that these experiments are performed and they would really add crucial information to our understanding of the role of osmotic stress in drug resistance. If these are outside the scope of the current manuscript the discussion should be provided with more nuance considering these factors.
Response: This is a very pertinent question. It is true that upregulation of certain genes, such as those encoding MDR efflux pumps, porins or specific drug resistance genes, caused by the ∆mchK mutation should, as indicated by the reviewer, increase the MICs of many antibiotics, including rifampicin. In fact, this was previously indicated in the manuscript. Nevertheless, it is difficult, although not impossible, to imagine how this upregulation could produce collateral susceptibility to many others. It would be interesting to know if expression of other genes, including efflux pumps and porins, are affected by mchK deletion. However, we consider that, as the reviewer insinuates in his/her comment, the study of gene expression is out of the scope of this paper and deserves to be characterized in a new study, as indicated in the Discussion.
In our manuscript, as indicated above, a paragraph related to this possibility was previously included in the Discussion section. This paragraph is now modified to highlight this possibility according to reviewer´s suggestion: “The antibiotic susceptibility profile of bacteria also depends on other factors, including porins and drug efflux pumps 29. For instance, the ABC-type multidrug efflux pumps can interact with the Trk system in E. coli 30. Moreover, the membrane potential and intracellular pH are important factors that regulate the activity of the drug efflux pumps depending on PMF in prokaryotes 31,32. The K+ transport could also have effects on the function and/or regulation of drug efflux pumps. Therefore, the effects of mchK inactivation on M. smegmatis antibiotic susceptibility could also be attributed, at least in part, to a differential regulation of other genes under osmotic stress, including efflux pumps and/or porins. The relationship between the K+ transport systems and the function of drug efflux pumps deserves to be studied in the future to identify additional mechanisms of drug resistance in mycobacteria 33”.
Comment #2
The second consideration is that there is a confusing disconnect between the first and second parts of the paper as it is written. The first part predominantly deals with drug resistance and the second part deals with characterizing mchK and its role in monovalent cation and pH balance. Within the scope of this journal the first part is more interesting.
The finding that increasing [K+] poses one very obvious experiment perhaps within the scope of this work is the impact of [K+] on the MIC of rifampicin for WT vs ∆mchK. The findings in the second part gain relevance if such an experiment is included. I recommend that the authors perform this experiment.
Otherwise the paper is well written and additional minor remarks are below.
Response: While it is clear that the paper has two differentiated parts, the second part is the study that allowed us to characterize the function of the mchK gene and to suggest one of the possible causes for the puzzling results of resistance: increased rifampicin resistance but collateral susceptibility to other antibiotics.
Following the timely reviewer´s suggestion, the study of the effect of [K+] on the MIC of rifampicin has been performed. However, variable, not concluding, results were obtained for both strains (WT and ∆mchK) when KCl was added to the growth medium in addition to rifampicin. We do not envisage the causes of these puzzling results (variation in growth medium?, different batch of antibiotics?, induction of other factors?…). As we were not able to obtain concluding results and in order to present honest conclusions, we have decided to soften the conclusion related to membrane potential as the exclusive cause of the Rif-R phenotype (it has been deleted from the title of the manuscript) and reinforce the alternative, but non-excluding, hypothesis of differential regulation of efflux pumps and or porins with new sentences (see previous response).
- Line 48
failure of chemotherapy treatment for TB.
Change to “antimicrobial chemotherapy” for TB, in order to disambiguate from cancer chemotherapy.
Response: Done.
- Line 71-72
The M. smegmatis target gene MSMEG_1945 identified in our screening was denominated mchK (for mycobacterial channel-K+).
Please also additionally denominate gene products MchKsm and MchKtb in the text.
Response: Denomination of these particular genes was previously included in the legend to Fig 1. However, following the reviewer suggestion, a new sentence has been included denominating gene products for M smegmatis and M tuberculosis. The new sentence: “Gene products were denominated MchKsm and MchKtb to define proteins from M. smegmatis and M. tuberculosis, respectively”.
- Line 75-81
The channel is formed by a homotetramer of MchK proteins, with each protomer consisting of two transmembrane helices and a pore domain in between them. While the channel domain could mediate the transport of K+ inside the cell, the RCK domain is considered to be the cytoplasmic sensor controlling the K+ gate 12–14 . This channel contains a selective filter inside the pore with the signature sequence TxGYG that indicates a specific high conductance of K+ ions.
These lines should be rewritten to reflect the fact that the authors are hypothesizing the mechanism and structure based on other K+ transporters. As written, it reads as though the cited references (12-14) are based on studies on the “MchK” protein.
Response: Thank you for the suggestion. To avoid confusion, the new paragraph includes the following changes in the sentences:
- “Based on its homology to other K+ transporters 12–14, the MchKsm protein is predicted to contain two putative domains:…”
- The channel, as in other K+ transporters, is putatively formed by a homotetramer of MchK proteins,
- Also, the last sentence of the paragraph “This channel sequence contains a selective filter inside the pore with the signature sequence TxGYG that indicates a specific high conductance of K+ ions”, has been changed to “The sequence TxGYG, inside the channel sequence, is the signature of a selective filter that indicates a specific high conductance of K+ ions 12–14”
- Line 95
Provide abbreviation expansion for MIC
Response: Done
- Line 286-296
Methods - Please provide antibiotic concentrations used.
Response: We deduce that reviewer is referring to concentrations used for studying survival rates. Antibiotic concentrations have been now included and the new sentence is: To measure survival rates in the presence of different concentrations of rifampicin (1, 2, 4 and 8 µg/ml), isoniazid (0.04, 0.08, 0.16 and 0.32 µg/ml) and ciprofloxacin (16, 32, 64 and 128 µg/ml).
Round 2
Reviewer 2 Report
It is disappointing that conclusive results did not arise from the [K+] experiment. The changes in conclusion and discussion adequately address the nuances of the various factors given the complications. The paper can be accepted as is.